# Probing the Association between Cognition, Suicidal Behavior and Tryptophan Metabolism in a Sample of Individuals Living with Bipolar Disorder: A Secondary Analysis

**DOI:** 10.3390/brainsci13040693

**Published:** 2023-04-20

**Authors:** Pasquale Paribello, Alessio Squassina, Claudia Pisanu, Anna Meloni, Stefano Dall’Acqua, Stefania Sut, Sofia Nasini, Antonella Bertazzo, Donatella Congiu, Mario Garzilli, Beatrice Guiso, Federico Suprani, Vittoria Pulcinelli, Maria Novella Iaselli, Ilaria Pinna, Giulia Somaini, Laura Arru, Carolina Corrias, Federica Pinna, Bernardo Carpiniello, Stefano Comai, Mirko Manchia

**Affiliations:** 1Section of Psychiatry, Department of Medical Sciences and Public Health, University of Cagliari, 09121 Cagliari, Italy; p.paribello@studenti.unica.it (P.P.); m.garzi@gmail.com (M.G.); beatrice.guiso@gmail.com (B.G.); federicosuprani@hotmail.it (F.S.); vittoriapulcinelli@hotmail.com (V.P.); novella.iaselli@gmail.com (M.N.I.); ilaria.pinna1991@gmail.com (I.P.); giulia444@alice.it (G.S.); carol.corrias@gmail.com (C.C.); fedepinna@inwind.it (F.P.); bcarpini@iol.it (B.C.); 2Unit of Clinical Psychiatry, University Hospital Agency of Cagliari, 09121 Cagliari, Italy; 3Department of Biomedical Science, Section of Neuroscience and Clinical Pharmacology, University of Cagliari, Monserrato, 09042 Cagliari, Italy; squassina@unica.it (A.S.); claudia.pisanu@unica.it (C.P.); anna.meloni@unica.it (A.M.); dcongiu@unica.it (D.C.); 4Department of Pharmaceutical and Pharmacological Sciences, University of Padova, 35131 Padova, Italy; stefano.dallacqua@unipd.it (S.D.); stefania.sut@unipd.it (S.S.); sofia.nasini@phd.unipd.it (S.N.); antonella.bertazzo@unipd.it (A.B.);; 5Department of Biomedical Sciences, University of Padova, 35131 Padova, Italy; 6San Raffaele Scientific Institute, 20132 Milano, Italy; 7Department of Psychiatry, McGill University, Montreal, QC H3A 1A1, Canada; 8Department of Pharmacology, Dalhousie University, Halifax, NS B3H 0A2, Canada

**Keywords:** hot cognition, affective recognition, suicide lifetime attempts, tryptophan metabolism

## Abstract

*Background and Objectives*: Alterations in hot cognition and in the tryptophan metabolism through serotonin (5-HT) and kynurenine (KYN) pathways have been associated with an increased risk of suicidal behavior. Here, we aim at probing the association between Stroop test performances and tryptophan pathway components in a sample of individuals with bipolar disorder (BD). *Materials and Methods*: We explored the association between the Emotion Inhibition Subtask (EIS) performances of the Brief Assessment of Cognition for Affective Disorders (BAC-A) and plasmatic levels of 5-hydroxytriptophan (5-HTP), 5-HT, KYN, 3-hydroxykynurenine (3-HK), quinolinic acid (QA), and kynurenic acid (KYNA) among subjects reporting lifetime suicide ideation (LSI) vs. non-LSI and subjects reporting lifetime suicide attempts (LSA) vs. non-LSA. *Results*: In a sample of 45 subjects with BD, we found a statistically significant different performance for LSA vs. non-LSA in the color naming (CN) and neutral words (NW) EIS subtasks. There was a significant association between CN performances and plasma 5-HTP levels among LSI and LSA subjects but not among non-LSI or non-LSA. *Conclusions*: In our sample, patients with LSA and LSI presented lower performances on some EIS subtasks compared to non-LSA and non-LSI. Moreover, we found an inverse correlation between plasma 5-HTP concentration and some EIS performances in LSA and LSI but not among non-LSA or non-LSI. This may represent an interesting avenue for future studies probing this complex association.

## 1. Introduction

### 1.1. Bipolar Disorder

Bipolar disorder (BD) is a complex and heterogeneous psychiatric condition characterized by the presence of severe comorbidities and a significant socioeconomic impact as it may represent up to 0.4% of the total DALY and 1.3% of the total YLD [1]. BD is associated with an average reduction in life expectancy of up to 20 years as compared to the general population [2], with certain lines of evidence suggesting that the gap in mortality may have further widened in more recent decades [3]. In the past, this excess mortality has been primarily attributed to suicide, but it has become increasingly evident how mortality from natural causes (determined by somatic diseases) may play a significant role with cardiovascular and oncological disorders being among the most frequent causes of death for this population [4,5]. 

### 1.2. Suicidal Behavior and Bipolar Disorder

The suicide rate for individuals affected by BD has been estimated to be 20–30 times that observed in the general population [6], underscoring its significance as a cause of morbidity and mortality for this population. Even though it is currently impossible to predict who is ultimately going to die by suicide, it is, however, possible to enact preventive strategies by managing risk factors for this outcome, akin to what cardiologists and general practitioners might do to prevent death by myocardial infarction [7]. Currently, suicide is conceptualized as a negative outcome in medicine resulting from a complex interplay of biological, psychological, social and environmental factors [7]. The weight of each singular component on the overall risk estimate may be difficult to disentangle and should instead be considered in the context of a dynamic and hierarchically complex interaction in which each component influences the remaining ones not necessarily in a proportionate manner. For example, genetic loading and environmental factors can each influence both the underlying biology (including cell functions, metaphysiology, neurophysiology) and the psychological traits (including but not limited to cognitive functions and behaviors). However, it is impossible to disentangle psychological traits from the underlying neurobiology itself, since the former is one of its many expressions [7]. Therefore, such persistent risk factors may interact with more proximal elements, such as stressful life events, in influencing the individual suicide risk profile at a particular time in the course of the illness [8]. 

### 1.3. Cognitive Functioning, TRP Metabolism and Suicide Risk

A growing body of evidence suggests that impaired cognitive functions may play a significant role in increasing the individual vulnerability to suicide [9,10,11,12]. Numerous cognitive tests have been assessed in this framework. Among them, the Stroop test presents an adequate amount of evidence supporting its significance in this setting [9,10,11]. Previous reports both in animal models [13] and in humans [14,15,16] aimed at modulating the serotonin (5-HT) system by either increasing the synthesis of 5-HT with 5-hydroxytryptophan (5-HTP) supplementation or reducing the brain levels of 5-HT using the acute tryptophan (TRP) depletion have suggested that variations in 5-HT levels may be associated with fluctuations in the level of attention. However, the evidence in this area of research is mixed, with other reports failing to find an association between acute TRP depletion and Stroop test performances [17]. TRP depletion has been associated with a differential effect on Stroop performances between individuals with a positive family history for mood disorders as compared to controls, leading to speculations about possible baseline anomalies in TRP metabolism associated with certain features of cognitive disfunction in psychiatric disorders [15]. However, few studies have investigated this association in patient populations. In general, low levels of 5-HT have been associated with higher impulsive aggression, defined as a disproportionate response to environmental stimuli, to either real or perceived threats [18]. For instance, forms of functional aggression—in other words, complex behaviors targeting at establishing control over a specific territory—have been linked to 5-HT function [19]. At this stage it is still unclear whether precursors of 5-HT may be associated with an increased risk of impulsive behavior, of either aggressive or suicide acts [20]. A report [21] focused on probing the association between TRP-kynurenine (KYN) pathway and cognition showed an association between a subscore of a verbal memory subtask and KYN metabolites levels among male BD patients but not among controls or female BD patients. Moreover, our group and others demonstrated an association between TRP metabolites levels in biological fluids and suicide risk in affective disorders [20,22,23]. However, evidence on a possible association between TRP metabolites, cognition and suicide risk in clinical samples is still to be clarified. Here, we present the results of a secondary analysis probing the potential association between cognitive functioning, suicidality [lifetime suicide attempts (LSA) and lifetime suicide ideation (LSI)], TRP metabolism along the KYN and 5-HT pathways in individuals with BD.

## 2. Materials and Methods

### 2.1. Study Sample

The study sample has been the subject of previous reports [24,25] and comprises 50 individuals affected by BD and 48 healthy controls (HC) recruited consecutively from 2019 to 2021 at the Unit of Psychiatry of the University of Cagliari and University Hospital Agency. This study aims to test for the association between genetics, gut microbiota, TRP metabolism through KYN and 5-HT with BD illness trajectories. The sample size was considered adequately powered to detect the presence of a difference between the study groups with an α set at 0.05 and an effect size of 5.3 [26]. The BD diagnosis was confirmed through the use of the Structured Clinical Interview for DSM-5 CV (SCID-5) [27]. The following inclusion criteria were applied: (1) a BD diagnosis, (2) older than 18 years old, (3) in euthymic state at the time of the recruitment as defined by a Hamilton Depression Rating Scale (HDRS) score <14 and a Young Mania Rating Scale (YMRS) score <13 points, respectively. The exclusion criteria were the following: (1) fertile age women with no contraception, (2) history of traumatic brain injury, (3) current or lifetime history of psychiatric or neurological disorders or other uncontrolled medical conditions, (4) current or lifetime history of substance use disorder, (5) treatment with melatonergic compounds (e.g., agomelatine, melatonin) within two months from recruitment. Study participants signed an informed consent form approved by the local independent ethical committee (Comitato Etico Indipendente dell’Azienda Ospedaliera Universitaria di Cagliari: PH/2019/6277) compliant with the most recent review of the Helsinki Declaration and the current EU legislation for privacy protection. At the time of recruitment, researchers collected an accurate psychiatric and medical history along with sociodemographic data. 

### 2.2. Recruitment Process

Study subjects were selected from a database of individuals living with BD and receiving treatment at the Unit of Psychiatry of the University of Cagliari. The first screening process involved analyzing the clinical health record available. Individual patients failing to satisfy inclusion and exclusion criteria mandated by the study protocol were excluded at this stage. Subsequently, selected patients were approached through their responsible clinician during routine assessments and were subsequently assessed by study investigators. Figure 1 summarizes the recruitment process.

### 2.3. Psychometric Assessments

Among the psychometric scales employed in the study [28,29,30,31,32], the Brief Assessment of Cognition in Affective Disorder (BAC-A) is of relevance here [32]. This scale is based on the Brief Assessment of Cognition in Schizophrenia (BACS) [33], a paper-and-pencil instrument that has been initially developed for schizophrenia patients and showed high correlation with other standard cognitive battery. It typically takes 35 min to complete and offers different versions to allow for repeated measurements over time [33]. BAC-A shares the first six subtests with the BAC-S (i.e., categorical and letter fluency, Tower of London, digit sequencing task, verbal memory and token motor task) with the addition of two further subtests: the affective interference tests (further divided into the affective processing and delayed recognition subtasks) and the emotion inhibition subtask (EIS). The EIS is a modified version for the Emotional Stroop task, in which participants are being shown four columns of either words or symbols and are given 30 s to read the word or name in the corresponding ink color as fast as possible [34]. The main objective for the EIS is to assess the capacity to suppress insignificant stimuli and read a word written with an ink color that may not match the meaning of the word itself (interference) [32]. The EIS is composed of four subtasks: (1) color naming (CN), where study subjects are asked to name the ink color for the listed symbols; (2) neutral color words, where study subjects are asked to name the color of the ink for the listed words with a neutral meaning; (3) affective color words, where study subjects are also asked to name the ink color for a list of words with affective meaning; (4) neutral words (NW), a task where study subjects are asked to read the words with a neutral meaning and written in black ink. The BAC-A has been validated in several languages and normative data exist for the Italian population [34]. A psychometric battery of tests has been employed to further characterize the clinical states of the study subject at recruitment, including the following: (1) HDRS [30], a clinician-rated instrument to assess the severity of depression, euthymia was defined as HDRS <c14; (2) YMRS [35], a clinician-rated instrument to assess the severity of mood elevations, euthymia was defined as YMRS <13; (3) Clinical Global Impression Severity [31] a clinician-rated instrument to evaluate the global severity; (4) Barratt Impulsiveness Scale [28], a self-report scale assessing impulsivity; (5) Hamilton Anxiety Rating Scale [29], a clinician-rated instrument to assess the severity of anxiety. The test panel was selected based on extensive literature supporting its use as well as on the need to provide an adequate severity assessment of the mood state. Study researchers received adequate training on the formal use of the said assessments. All psychometric tools employed for this study were available in the Italian language.

### 2.4. Laboratory Analysis

The employed laboratory analyses have been described in previous reports [24,25,36,37,38]. Blood samples were gathered in the early morning for each of the recruited subjects with EDTA tubes, immediately centrifuged at 2500 rpm at 4 °C for 10 min. Plasma aliquots were then separated and stored at −80 °C. Within 4 months from the time of collection, the plasma levels of TRP, 5-HTP, 5-HT and KYN were assessed through the use of HPLC system employing UV–Vis and fluorometric detectors, whilst quinolinic acid (QA), kynurenic acid (KYNA), and 3-hydroxykynurenine (3-HK) plasma levels were assessed through LC-MS/MS with the use of alfa-methyltryptophan as an internal standard. The KYN/TRP*1000 ratio was calculated as a proxy of the indoleamine 2,3-dioxygenase and tryptophan-2,3-dioxygenase enzymatic activities, the 5-HTP/TRP ratio as a proxy of the activity of the tryptophan 5-hydroxylase activity, and the QA/KYNA ratio as a measure of neurotoxicity (NMDA agonism/NMDA antagonism).

### 2.5. Statistical Analysis

For statistical analyses, we employed the following tests: (1) *t*-test for parametric data, the Wilcoxon/Mann–Whitney test for non-parametric data, chi-square for categorical data; (2) linear regression to probe the association between different concentrations of TRP metabolites in HC, LSA vs. non-LSA; (3) linear regression to probe the association of EIS and plasmatic levels of, KYN, 5-HTP or 5-HT in LSA vs. non-LSA. A linear regression model was developed to evaluate the possible influence of total duration of illness and gender on the observed associations. Corrections according to the normative Italian population have been applied to the gathered raw data for the BAC-A performances [34]. All statistical analyses have been performed with RStudio [39] or JASP [40]. 

## 3. Results

### 3.1. Plasma TRP Metabolites BD vs. Controls Comparison

For the analysis focusing on the association of metabolite levels in the different study groups, we included 48 healthy controls (HC) and 50 individuals living with BD, of which 33 non-LSA and 17 with LSA. Table 1 summarizes the main features of the recruited sample, divided in the three different subgroups. No significant difference emerged in terms of gender or age. However, there was a statistically significant greater number of smokers or ex-smokers in non-LSA compared to HC (chi-square, *p* < 0.001). Further, we found a lower number of individuals practicing physical activity in non-LSA compared to HC (chi square, *p* = 0.003). A significantly greater concentration of TRP emerged in HC as compared to both LSA (Wilcoxon, *p* = 0.006) and non-LSA (Wilcoxon, <0.001), but greater 5-HTP concentration among both LSA (Wilcoxon, *p* = 0.0059) and non-LSA vs. HC (Wilcoxon, *p* = 0.001). In addition, the 5-HTP/TRP ratio resulted higher among LSA and non-LSA as compared to HC. Quinolinic acid concentration appeared significantly higher among HC as compared to non-LSA (Wilcoxon, *p* = 0.015) but not with LSA. No significant difference emerged between LSA vs. non-LSA for any of the other tested metabolites. Moreover, no statistically significant difference emerged for the remaining metabolites or the tested ratios for the comparison HC vs. non-LSA or HC vs. LSA. BMI levels resulted higher among non-LSA compared to HC (Wilcoxon, *p* = 0.04), but the difference was not significant for the LSA subgroup.

Considering the observed difference of TRP and 5-HTP among non-LSA and LSA compared to HC, further analyses were performed to test for a differential pattern of TRP metabolites levels between the different study groups. A linear model was applied to probe the association of 5-HTP and TRP plasma concentrations in LSA, non-LSA and HC, but the tested association was statistically significant only for the LSA subgroup (R = 0.5, *p* = 0.043—Panel A Figure 2). Even when correcting for gender, age, BMI and physical activity in a linear regression model, 5-HTP concentration (*p* = 0.040) and the presence of lifetime suicide attempts (<0.001) remained significantly associated with TRP concentration. However, the overall model was significant only when adding the presence of LSA as a cofactor. Similarly, a significant association emerged for the association between KYN concentration and TRP but only for HC (R = 0.31, *p* = 0.029—Panel B Figure 1). The association resulted AS no longer statistically significant in a linear regression model when correcting for the presence of lifetime suicide attempts, physical activity, age, gender and BMI. 

### 3.2. Comparison of Cognitive Performances of LSA vs. Non-LSA 

For the analyses of the association between circulating concentrations of TRP metabolites and cognitive performances, we considered only the 45 BD patients who completed the cognitive tests. The sample composition and the distribution (mean, median and SD) of the EIS is described in Table 2. The most prevalent diagnosis was BD type I (*n* = 32, 71.1%), followed by BD type II (*n* = 13, 28.8%). The average age at onset was 26.5 years, and the average disease duration at recruitment was 28.4 years. 

Additional information on disease duration and number of episodes divided among LSA vs. non LSA is summarized in Table 3.

At the time of recruitment, 43 out of 45 subjects were prescribed a pharmacological therapy. No statistically significant difference emerged in terms of pharmacological therapy (antipsychotics, antidepressants, or mood stabilizers) between LSA vs. non-LSA or LSI vs. non-LSI (*p* > 0.05 for each comparison). In addition, there was no statistically significant difference in the plasma 5-HTP concentrations either between LSI vs. non-LSI (Wilcoxon, *p* = 0.9) or between LSA vs. non-LSA (Wilcoxon, *p* = 0.96). Similarly, no statistically significant difference emerged for HDRS-17 scores (Mann–Whitney, *p* = 0.221) or YMRS (Mann–Whitney, *p* = 0.565). CGI-S scores were significantly higher between LSI vs. non-LSI (*p* = 0.014) but not between LSA vs. non-LSA (*p* = 0.055). A statistically significant association was observed between the color naming subtask of the EIS (CN) performances and plasmatic 5-HTP for LSA but not for non-LSA (panel A Figure 3, R = −0.5, *p* = 0.049). Similarly, among LSI, an association was found for CN and 5-HTP, but not among non-LSI (panel B Figure 3, R = −0.52, *p* = 0.013). Moreover, a significant difference emerged between LSA vs. non-LSA, with the former presenting lower performances compared to the latter for the CN (panel C, Figure 3, Wilcoxon, *p* = 0.031) and the neutral words subtask of the EIS (NW, panel D, Figure 3, Wilcoxon, *p* = 0.0017) subtasks. For the LSI vs. non-LSI comparison, only for the NW subtask (Wilcoxon, *p* = 0.003) it was possible to observe statistically significant lower performances among LSI but not for the CN subtask (Wilcoxon, *p* = 0.052). To correct the observed association for possible confounders, we fitted a linear regression model to predict the amount of variation in the 5-HTP plasma concentrations that could be predicted by the CN and NW performances, with gender as a cofactor and total time of illness at recruitment (months) as covariate. When correcting for the total time of illness and gender, only the CN (*p* = 0.002) performances remained significantly associated with plasma 5-HTP concentrations but not the NW performances (*p* = 0.108). Neither gender (*p* = 0.608) nor the total time of illness (*p* = 0.555) appeared significantly associated with the dependent variable. 

## 4. Discussion

### 4.1. Suicide as a Target for Prevention 

Suicide represents a severe problem in public health and a significant cause of mortality in the general population. Stratification of suicide risk could allow clinicians to increase the accuracy of their therapeutic interventions on those individuals at highest risk. Compared to other medical fields, psychiatry might have focused too much on the limited accuracy of short-term prediction (i.e., weeks), underestimating the importance of having a reasonably efficient way to gauge suicide risk in the long term. This would be essential for the development of long-term management plans (i.e., months or years). Arguably, cardiologists employ risk stratification tools in their daily clinical practice by selecting those individuals that might benefit the most from specific treatments (i.e., statins). However, they might not necessarily be able to identify those that will ultimately die by myocardial infarction among individuals judged to be at risk (a rare event in a 12-month risk horizon, similarly to what is death by suicide in individuals with positive risk factors for suicide) [36]. Albeit it might not be possible to make accurate short-term predictions for any given patient for such a rare outcome, the possibility of streamlining treatment paradigms may still be valuable on a large scale and over longer periods of time [7]. A similar approach could be applied for suicide prevention. In the context of a precision medicine approach, identifying subgroups of individuals through the integration of clinical, sociodemographic, multi-omics and neuroimaging data could allow us to better focus our resources where they are most needed with the aim of aggressively addressing risk factors for suicide [37]. Machine learning approaches applied to the analysis of sociodemographic and clinical determinants of suicide risk, among the others, have yielded models that might be beneficial in this sense and may be applied to psychiatric inpatients in the post-discharge treatment planning [36]. Risk profiling of patients based on their clinical history is typically applied in clinical practice by identifying elements such as previous history of suicide attempts, violent attempts, a high genetic loading, and early childhood traumas such as sexual abuse. The evaluation of cognition could support the assessment of the individual suicide risk profile [38,41], and in a dimensional and transdiagnostic framework, exploring the association between TRP metabolism and cognition could be beneficial both in informing new diagnostic-treatment protocols and in improving our understanding of the underlying mechanisms. 

### 4.2. Preliminary Results from Our Cohort: Possible Future Directions

In our sample, LSA subjects presented an apparently higher conversion of TRP to 5-HTP and worse performances in some EIS tests compared to non-LSA, and it was also possible to determine an inverse correlation between plasma 5-HTP levels and cognition. It is possible to observe a similar tendency for the 5-HTP levels and cognition, albeit non-significant, in the overall sample. Our results should be considered preliminary, as it was not possible to establish any causal link between the study variables considering the study design and the limited sample size. However, it is not unrealistic to hypothesize that LSA subjects might present a greater susceptibility to variations in plasma levels of TRP metabolites, and that this phenomenon could contribute to worse cognitive performances. Some lines of evidence suggest that TRP metabolism could be influenced by the gut microbiota composition, adding a further layer of complexity to the interaction between cognition, suicidality and TRP metabolism [42]. A relatively untapped area of future research might be represented by the assessment of the oral microbiota composition [43,44] since at least one report suggests a possible association between saliva microbiota composition and suicidal ideation [45]. Arguably, sampling the saliva microbiota could be easier as compared to the gut microbiota and thus more suitable for the potential development of clinically expendable biomarkers. Future studies could, for example, evaluate the efficacy of biomarkers assessing the oral microbiota in predicting the probability of response to pharmacological or non-pharmacological treatment interventions for individuals being admitted to an acute inpatient unit for an attempted suicide and how these biomarkers could correlate with cognition. In this setting, evaluating the potential association between such elements, cognition and multi-omics in adequate prospective studies could further improve our knowledge on this complex interaction and potentially inform new diagnostic and treatment algorithms.

### 4.3. Study Limitations

The results of this study should be interpreted in the context of several limitations. First, the sample size was originally calculated to answer a different research question, leading to the selection of a relatively small sample size for this secondary analysis. The original project did not include cognitive assessment for healthy controls, further reducing the amount of available data points. Therefore, the findings from this report should only be considered preliminary. In addition, due to the limited sample size, in the presented model no specific correction was performed based on other confounders (e.g., BD subtype, pharmacological therapy), possibly leading to a further reduction in its external validity. Albeit subject of previous reports, at this stage, the biological or clinical significance of the association between 5-HTP concentrations and cognitive performances remains uncertain. For the comparison of TRP metabolites between either non-LSA or LSA with HC, it was impossible to correct for the possible effects of pharmacological treatments. We did not find a significant difference between LSA vs. non-LSA groups in terms of symptom severity; therefore, at this stage, we do not consider this variable as a significant confounding factor for our analysis. Nonetheless, we cannot exclude that the cognitive performances and TRP metabolite levels may change significantly during acute mood episodes or during acute suicide crises, further complicating the overall interpretation of the interplay between TRP metabolism and cognition. Ultimately, our results should be considered preliminary, deserving confirmation in larger cohorts, further assessed through multiomics analysis to probe the complex interplay of TRP metabolism and cognition. Moreover, a transdiagnostic approach rather than focusing on a single diagnostic category may improve our understanding of the underlying neurobiological mechanisms at play. However, the observation of an apparently different level of correlation between TRP and 5-HTP levels between LSA vs. non-LSA may still suggest the involvement of this metabolic pathway, as there was no significant qualitative difference in the prescribed medication regimen.

## 5. Conclusions

Suicidal behavior represents an adverse clinical outcome deriving from a complex interaction of interdependent biological, neuropsychological, and socio-environmental factors. Our data suggest that cognition and activity of TRP metabolism, particularly the levels of 5-HTP, the precursor of the neurotransmitter 5-HT, could be involved in modulating suicide risk. This interaction may represent one of the many mechanisms underlying a higher vulnerability to suicide.

## Figures and Tables

**Figure 1 brainsci-13-00693-f001:**
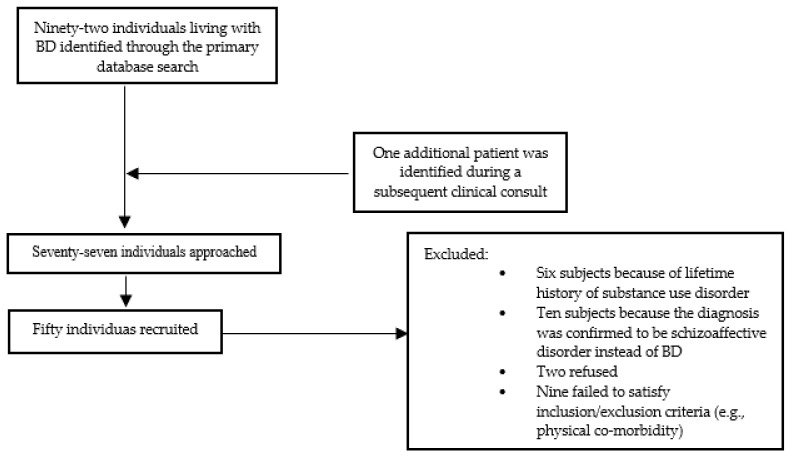
Summary of the recruitment process.

**Figure 2 brainsci-13-00693-f002:**
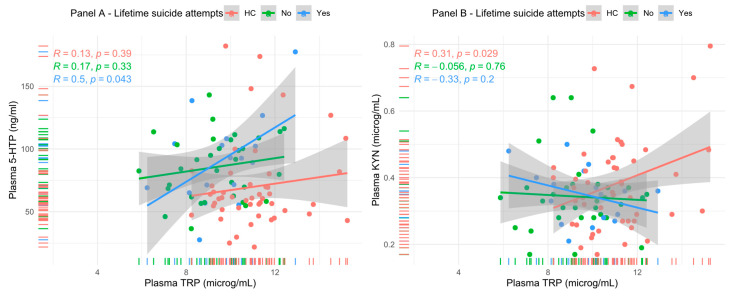
Relationship between plasma TRP and two of its major metabolites, 5-HTP and KYN.

**Figure 3 brainsci-13-00693-f003:**
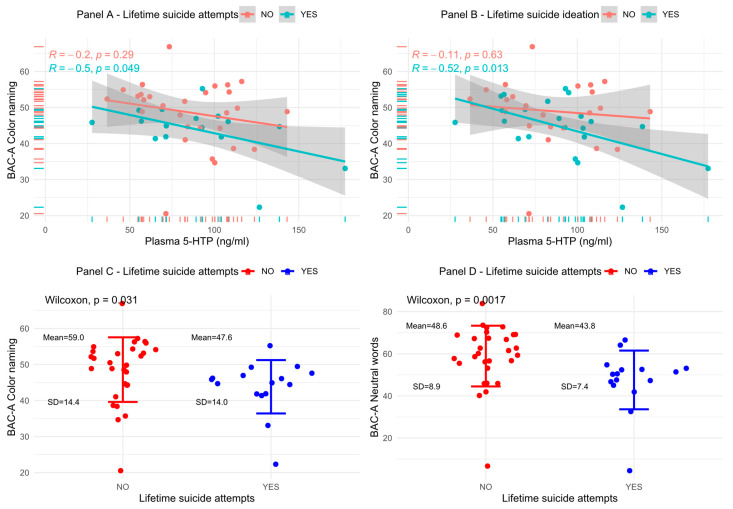
Relationship between cognitive measures, lifetime suicide behavior and plasma 5-HTP levels.

**Table 1 brainsci-13-00693-t001:** Sociodemographic data and plasma concentration for the tested TRP metabolites.

	Healthy Controls (*n* = 48)	Non-LSA (*n* = 33)	LSA (*n* = 17)	LSA vs. Non-LSA	HC vs. LSA	HC vs. Non-LSA
Gender—n (%)						
Female	30 (62)	19 (57)	12 (70)	0.3 ^§^	0.5 ^§^	0.6 ^§^
Male	18 (37)	14 (42)	5 (29)			
Physical activity—n (%)						
No	19 (39)	24 (72)	11 (64)	0.55 ^§^	0.07 ^§^	0.003 ^§^
Yes	29 (60)	9 (27)	6 (35)			
Cigarette smoking—n (%)						
No	33 (69)	9 (27)	4 (23)	0.5 ^§^	1.0 ^§^	<0.001 ^§^
Yes	9 (19)	14 (42)	10 (59)			
Ex-smoker	6 (12)	9 (27)	3 (18)			
BMI mean-median-SD	23.6-23.3-3.4	26.4-25.0-6.2	24.6-25.5-4.0	0.4 ^^^	0.2 ^^^	0.04 ^^^
Age at T0—years (mean-median-SD)	52-53-8	51-53-8.5	50-49-8.8	0.33 ^^^	0.42 ^^^	0.31 ^^^
TRP (μg/mL) mean-median-SD	11.0-10.9-1.7	9.4-9.8-1.6	9.6-9.8-1.6	0.7 ^^^	0.0065 ^^^	<0.001 ^^^
5-HTP (ng/mL) mean-median-SD	69.9-60.7-34.4	85.7-84.2-25.4	91.2-92.6-35.4	0.7 ^^^	0.0059 ^^^	0.001 ^^^
5-HT (ng/mL) mean-median-SD	260.7-241.9-167.9	288.7-269.9-141.2	363.3-380.1-204.8	0.29 ^^^	0.063 ^^^	0.28 ^^^
KYN (μg/mL) mean-median-SD	0.383-0.376-0.140	0.343-0.330-0.109	0.350-0.360-0.083	0.53 ^^^	0.53 ^^^	0.17 ^^^
3-HK (ng/mL) mean-median-SD	44.4-42.4-13.5	42.8-42.2-3.2	41.9-41.8-1.6	0.36 ^^^	0.34 ^^^	0.65 ^^^
QA (ng/mL) mean-median-SD	158.7-154.5-45.9	141.2-142.6-14.5	145.6-141.2-15.7	0.6 ^^^	0.2 ^^^	0.015 ^^^
KYNA (ng/mL) mean-median-SD	10.2-8.3-8.0	8.3-8.4-4.1	10.2-9.9-5.5	0.13 ^^^	0.48 ^^^	0.67 ^^^
5-HTP/TRP*1000 ratio mean-median-SD	6.4-5.7-3.2	9.3-9.5-3.1	9.5-9.2-3.3	0.8 ^^^	<0.001 ^^^	<0.001 ^^^
KYN/TRP*1000 ratio mean-median-SD	34.8-34.0-11.9	37.8-32.9-14.3	38.0-33.7-14.4	0.9 ^^^	0.47 ^^^	0.6 ^^^
QA/KYNA ratio mean-median-SD	22.5-20.7-14.0	23.1-17.4-17.8	24.9-13.6-25.6	0.17 ^^^	0.28 ^^^	0.8 ^^^

Abbreviations: HC—healthy controls; LSA—Lifetime suicide attempts; non-LSA—no Lifetime suicide attempts; TRP—tryptophan; 5-HTP— 5-hydroxytryptophan; 5-HT—serotonin; 3-HK—3-hydroxykynurenine; QA— quinolinic acid; KYNA—kynurenic acid; ^§^ χ^2^ *p* value; ^^^ Wilcoxon test *p* value.

**Table 2 brainsci-13-00693-t002:** Sociodemographic data and psychometric test results for the sample of 45 patients with BD.

Variable	Females	Males	χ^2^/Mann–Whitney *p*
Gender—n (%)	30 (67)	15 (33)	0.36
Socioeconomic level—n (%)			
Low	11 (37)	5 (33)	0.95
Average	11 (37)	6 (40)	
High	8 (26)	4 (27)	
Marital status—n (%)			
Single	8 (27)	7 (47)	0.21
Married	5 (17)	4 (32)	
Divorced	11 (37)	3 (16)	
Widow	1 (3)	0 (0)	
In stable relationship	5 (16)	1 (6)	
Education—n (%)			
≤11 years	7 (23)	6 (40)	0.18
11–13 years	14 (47)	7 (47)	
+13 years	9 (30)	2 (13)	
Lifetime history of suicide ideation (LSI)—n (%)			
No	13 (43)	10 (67)	0.10
Yes	17 (57)	5 (33)	
Lifetime history of attempted suicide (LSA)—n (%)			
No	18 (60)	11 (73)	0.36
Yes	12 (40)	4 (27)	
HDRS total-median score	5.0	2.0	0.19
YMRS total-median score	0	0	0.45
CGI-S-median score	2	3	0.043
BIS-median score	69	68	0.21
BAC-A-EIS 45 subjects, (%) subsample with performances in the pathological range			
Color naming, mean-median and SD (% abnormal results-equivalent score-ES 0)	46.9-48.4-9.7 (13.3)	46.9-46.8-6.3 (6.6)	0.78
Neutral color, mean-median score and SD (% abnormal results-equivalent score-ES 0)	41.9-44.1-8.7 (13.3)	42.0-42.2-5.1 (0.0)	0.27
Affective color words, mean-median score and SD (% abnormal results-equivalent score-ES 0)	40.2-41.8-9.4 (16.6)	42.2-42.8-4.8 (6.6)	0.43
Neutral words-mean, median and SD (% abnormal results-equivalent score-ES 0)	52.3-55.1-15.8 (13.3)	60.1-62.7-12.4 (13.3)	0.20

Abbreviations: BAC-A—Brief Assessment of Cognition for Affective disorders; BIS—Barratt Impulsive Scale; EIS—Emotion Inhibition Subtask; *p*—*p*-value.

**Table 3 brainsci-13-00693-t003:** Total disease duration and number of mood episodes in LSA vs. non-LSA.

	LSA		χ^2^/Mann–Whitney p
Family history of suicide attempts (FHSA)	No	Yes	
No	26	14	0.76
Yes	7	3	
Total duration of illness (months, median)	24	28	0.24
Total time spent in major depressive episode (months, median)	16	21	0.54
Total time spent in (hypo)mania (months, median)	6	9	0.86
Number of total (hypo)mania episodes (median)	4	4	0.78
Number for total major depressive episode (median)	3	6	0.039
Age at first attempt (y.o.)			
Median	31.0		
Mean	34.0		
Youngest at 1st attempt	21.0		
Oldest at 1st attempt	53.0		
Polarity of mood at 1st attempt (n = 17)			
Mania	2		
Major Depressive Episode	11		
Mixed	7		

## Data Availability

Study data are available upon request to the corresponding authors.

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
