# Peer review of "Probing the Association between Cognition, Suicidal Behavior and Tryptophan Metabolism in a Sample of Individuals Living with Bipolar Disorder: A Secondary Analysis"

_brainsci, 2023, doi:10.3390/brainsci13040693_

Round 1

Reviewer 1 Report

The aim of this study was to investigate the relationship between suicide and cognitive elements (stroop test) and tryptophan pathway components in patients with bipolar disorder.  In this process, various psychometric tests and blood levels of tryptophan pathway components (tryptophan, serotonin, kynurenine) were evaluated. The study was conducted on 45 bipolar patients and 46 controls, and the results of the Emotion Inhibition Subtask and secondary analyses revealed that suicide attempt had some relationship with 5HTP concentrations. The article is within the scope of the journal and will contribute to the literature. Some of my suggestions about the study are as follows;

1. The fact that the paragraphs are not used much in the article has seriously affected its comprehensibility and flow, the article should be reorganized in this respect.

2. In this context, subheadings and sections should be created in the methodology, the process of including the cases in the study and how many people were excluded and why should be explained.

3. Similarly, the purpose of the scales used and their psychometric properties in both the original and the language used should be explained.

4. It would be useful to include more information about the processes and clinical features of bipolar disorder. In particular, it would be useful to include the number of previous episodes, their characteristics and their relationship with variables. 

5. The timing of the blood values taken, how they were taken and stored, and how they were evaluated should be given. Since this study was a secondary analysis of a previous study, this information may have been given in the previous study. Nevertheless, the authors should provide brief information on this subject and explain how the examinations were performed and at which stage or during which attack period. 

6. The findings section should be organized in a more understandable way and divided into groups.

7. The readability of the discussion is very difficult. There should be a discussion of how these data may change, especially in different episodes.  

Author Response

The aim of this study was to investigate the relationship between suicide and cognitive elements (stroop test) and tryptophan pathway components in patients with bipolar disorder. In this process, various psychometric tests and blood levels of tryptophan pathway components (tryptophan, serotonin, kynurenine) were evaluated. The study was conducted on 45 bipolar patients and 46 controls, and the results of the Emotion Inhibition Subtask and secondary analyses revealed that suicide attempt had some relationship with 5HTP concentrations. The article is within the scope of the journal and will contribute to the literature. Some of my suggestions about the study are as follows.

Q1) The fact that the paragraphs are not used much in the article has seriously affected its comprehensibility and flow, the article should be reorganized in this respect.

R1) Thank you for your feedback. We have reorganized the manuscript and additional paragraphs have been added to improve the drat readability.

Q2) In this context, subheadings and sections should be created in the methodology, the process of including the cases in the study and how many people were excluded and why should be explained.

R2) Additional subheadings and an additional image describing the recruitment process has been added.

Q3) Similarly, the purpose of the scales used and their psychometric properties in both the original and the language used should be explained.

R3) An additional description and references have been added for each of the employed psychometric tests

Q4) It would be useful to include more information about the processes and clinical features of bipolar disorder. In particular, it would be useful to include the number of previous episodes, their characteristics and their relationship with variables.

R4) An additional table was added describing the number of (hypo)mania and major depressive episodes (MDE) for individuals with lifetime suicide attempts vs non-lifetime suicide attempts, along with the total time of illness and total time spent in either MDE/ (hypo)mania episode.

Q5) The timing of the blood values taken, how they were taken and stored, and how they were evaluated should be given. Since this study was a secondary analysis of a previous study, this information may have been given in the previous study. Nevertheless, the authors should provide brief information on this subject and explain how the examinations were performed and at which stage or during which attack period.

R5) The following passage was added entitled -2.5 Laboratory analysis- “The employed laboratory analyses have been described in previous reports [24, 25, 38-40]. Blood samples were gathered in the early morning for each of the recruited subject with EDTA tubes, immediately centrifuged at 2500 rpm at 4 °C for 10 min. Plasma aliquotes were then separated and stored at −80 °C. Within 4 months from the time of collection, the plasma levels of TRP, 5-hydroxytryptophan, 5-HT, and KYN were assessed through the use of HPLC system employing UV–Vis and fluorometric detectors, whilst QA, KYNA, 3-HK, and MLT plasma levels were assessed through LC-MS/MS with the use of alfa-methyltryptophan as an internal standard.”

Q6) The findings section should be organized in a more understandable way and divided into groups.

R6) Additional subheadings have been added in the methods, results and discussion sections to improve the readability of the draft.

Q7) The readability of the discussion is very difficult. There should be a discussion of how these data may change, especially in different episodes. 
R7) The following passage was added to the Discussion to better address possible concerns regarding instability of either cognitive performances or TRP metabolism: “We did not find a significant difference between LSA vs non-LSA groups in terms of symptom severity, therefore at this stage we do not consider symptoms severity as a significant confounding factor for our analysis. Nonetheless, we cannot exclude that the cognitive performances and TRP metabolite levels may change significantly during acute mood episodes or during acute suicide crises, further complicating the overall interpretation of the interplay between TRP metabolism and cognition at the level of the individual patient. Ultimately, our results should be considered preliminary, deserving confirmation in larger cohorts, and further assessed through multiomics, specifically probing the complex interplay of TRP metabolism and cognition. Moreover, a transdiagnostic approach rather than focusing on a single diagnostic category may improve our understanding for the underlying mechanisms at play.”

Reviewer 2 Report

The authors discuss an important issue concerning the pathogenesis of bipolar affective disorder, mainly in the context of suicidal attempts.

The study covered a small group of patients and concerned only one disease entity.

Hence, the large number of authors of the work is puzzling.

I would describe the discussed results as preliminary, even if other research results were presented in this group. The small number of patients rather indicates a certain prelude to broader research.

When did the suicide attempts take place, at what stage of the illness, was such an interview collected?

  Were the respondents in a depressive or hypomanic or manic phase or perhaps in remission during the

In the introduction, I propose to discuss the role of tryptophan in impulsive behavior in more detail, because it plays a significant role in making the decision to commit suicide in various mental disorders. The reader may not see the cause-and-effect relationship within the meaning of the methodology of the presented study. That is why the authors linked these psychological tests with specific biological markers.

People with BD should be compared to other disease entities in terms of the analyzed variables in future research. Or raise the issue in the discussion.The authors discuss an important issue concerning the pathogenesis of bipolar affective disorder, mainly in the context of suicidal attempts.

Author Response

The authors discuss an important issue concerning the pathogenesis of bipolar affective disorder, mainly in the context of suicidal attempts. The study covered a small group of patients and concerned only one disease entity.

Q1) Hence, the large number of authors of the work is puzzling.

R1) We thank the reviewer for this observation. We would like to highlight that the study is based on a longitudinal prospective observation of this cohort of patients for two years. Hence, the clinical arm represented a substantial portion of the study, with monthly assessment of mood variation and eventual evaluation of affective and cognitive symptoms in case of a recurrence. This resulted in the commitment of several clinicians that contributed substantially to the clinical accurate characterization of these patients and therefore meet the criteria for authorship.

Q2) I would describe the discussed results as preliminary, even if other research results were presented in this group. The small number of patients rather indicates a certain prelude to broader research.

R2) The manuscript has been edited by adding the following passage: “Ultimately, our results should be considered preliminary, deserving confirmation in larger cohorts, further assessed through multiomics analysis to probe the complex interplay of TRP metabolism and cognition. Moreover, a transdiagnostic approach rather than focusing on a single diagnostic category may improve our understanding of the underlying neurobiological mechanisms at play.”

Q3) When did the suicide attempts take place, at what stage of the illness, was such an interview collected?

R3) We have added data regarding polarity and age at first attempt, along with the number of individuals with previous suicide attempt with positive family history of either attempted or completed suicide attempt.

Q4) Were the respondents in a depressive or hypomanic or manic phase or perhaps in remission during the

R4) The study protocol for the original study mandated euthymic state as an inclusion criterion, as defined according to the included psychometric test battery. To further clarify this element, we have added the following inclusion criteria in the corresponding section “3) in euthymic state at the time of the recruitment as defined with a HDRS <14 and YMRS < 13 points, respectively.”

Q5) In the introduction, I propose to discuss the role of tryptophan in impulsive behavior in more detail, because it plays a significant role in making the decision to commit suicide in various mental disorders. The reader may not see the cause-and-effect relationship within the meaning of the methodology of the presented study. That is why the authors linked these psychological tests with specific biological markers.

R5) The following paragraph has been added to the introduction section “In general, low levels of 5-HT have been associated with higher impulsive aggression, defined as a disproportionate response to environmental stimuli, to either real or perceived threats [22]. For instance, forms of functional aggression, that are complex behaviors targeting at establishing control over a specific territory, have been linked to 5-HT function [23]. At this stage it is still unclear whether precursors of 5-HT may be associated with an increased risk of impulsive behavior, of either aggressive or suicide acts [21].”

Q6) People with BD should be compared to other disease entities in terms of the analyzed variables in future research. Or raise the issue in the discussion. The authors discuss an important issue concerning the pathogenesis of bipolar affective disorder, mainly in the context of suicidal attempts.

R6) The following passage was added to the discussion section “We did not find a significant difference between LSA vs non-LSA groups in terms of symptom severity, therefore at this stage we do not consider this variable as a significant confounding factor for our analysis. Nonetheless, we cannot exclude that the cognitive performances and TRP metabolite levels may change significantly during acute mood episodes or during acute suicide crises, further complicating the overall interpretation of the interplay between TRP metabolism and cognition. Ultimately, our results should be considered preliminary, deserving confirmation in larger cohorts, further assessed through multiomics analysis to probe the complex interplay of TRP metabolism and cognition. Moreover, a transdiagnostic approach rather than focusing on a single diagnostic category may improve our understanding of the underlying neurobiological mechanisms at play.”